# Online Inverse Reinforcement Learning with Learned Observation Model

**Saurabh Arora & Prashant Doshi**
THINC Lab, Dept. of Computer Science
University of Georgia, Athens, GA
United States of America
{sa08751, pdoshi}@uga.edu

**Bikramjit Banerjee**
School of CSCE
Univ. Southern Mississippi, Hattiesburg, MS
United States of America
Bikramjit.Banerjee@usm.edu

**Abstract:** With the motivation of extending incremental inverse reinforcement learning (I2RL) to real-world robotics applications with noisy observations as well as an unknown observation model, we introduce a new method (RIMEO) that approximates the observation model in order to best estimate the noise-free ground truth underlying the observations. It learns a maximum entropy distribution over the observation features governing the perception process, and then uses the inferred observation model to learn the reward function. Experimental evaluation is performed in two robotics tasks: (1) post-harvest vegetable sorting with a Sawyer arm based on human demonstration, and (2) breaching a perimeter patrol by two Turtlebots. Our experiments reveal that RIMEO learns a more accurate policy compared to (a) a state-of-the-art IRL method that does not directly learn an observation model, and (b) a custom baseline that learns a less sophisticated observation model. Furthermore, we show that RIMEO admits formal guarantees of monotonic convergence and a sample complexity bound.

**Keywords:** Observation model, Inverse reinforcement learning, Maximum entropy

## 1 Introduction

Inverse reinforcement learning (IRL) methods learn the specification for a task that best explains the observed behavior of an expert executing that task [1, 2], in the form of a reward function for a Markov decision process [3]. IRL methods generally assume that the learner's perception is noise-free and that the learning occurs offline. However, real-world sensing is often noisy and agents may need to learn incrementally because not all the training data is available in one batch. Motivated by the challenging goal of bringing robotic automation to post-harvest processing lines for vegetables, we consider examples of noisy perception by a robot learner observing a human sorting onions by removing the blemished ones. To learn this complicated manipulation task from its observations, the learner needs to unambiguously understand when is an onion considered blemished, and how do the various locations of the expert's hand help realize the task (e.g., hand hovering over table, hand near eyes of the sorter); see Fig. 1(*a*). When the expert places a blemished onion in a bin for bad onions, as the robot's sensing is imperfect, the blemish may not be perceived correctly by the learner's camera. This introduces an error in the perceived state because the learner incorrectly sees the human placing an unblemished onion in a destination meant for bad onions (see Fig. 1(*b*) top). Another example of imperfect perception is shown in Fig. 1(*b*) bottom: although the expert is holding an onion just above the table, the learner may incorrectly perceive it to be in the inspection (or eye) region. This ambiguity occurs because these two regions are in close proximity.

Shahryari and Doshi [4] approach the problem of offline IRL under noisy observation using the *Robust IRL* method, but they assume that the learner knows the observation (or sensor) model. This assumption may not be pragmatic because the observation model is typically a function of obscure variables including inaccuracies in sensing hardware and computer vision. Real-world robotic

6th Conference on Robot Learning (CoRL 2022), Auckland, New Zealand.

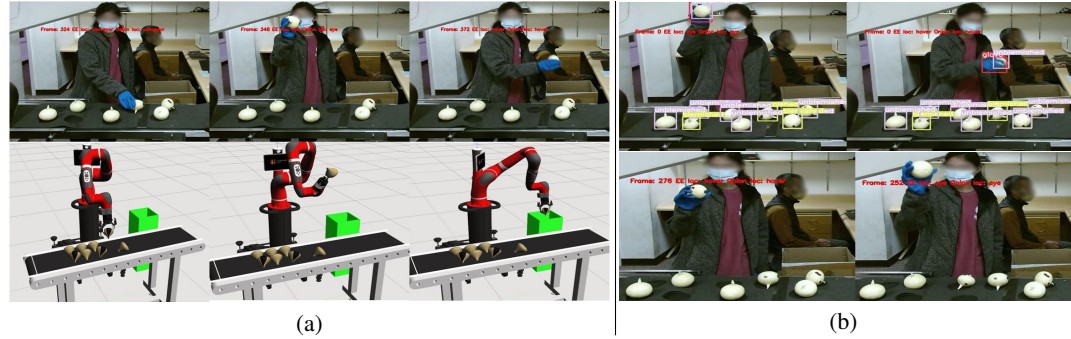

| (a) | (b) |

Figure 1: **(a)** (top) A human demonstrator picks an onion, inspects it closely to check if it is blemished, and places it in the bin on finding it to be blemished. (bottom) The learner (Sawyer) robot uses the learned policy to imitate the observed behavior. **(b)** (top) A blemished onion detected as being unblemished. (bottom-left) An onion held in the hover region. (bottom-right) An onion held at the eye (or in inspection) region.

applications need a method that can learn the unknown observation model under noisy input and use that model to learn a reward function.

When a learner passively watches the expert perform the task, the observations arrive as a continuous stream of data over an extended period of time. However, methods such as Robust IRL require all observations to be available together as a batch. Current IRL applications need a method that can learn *online* from incremental noisy input without a'priori knowledge of the sensor model. To meet this challenge, we present a method that generalizes the batch Robust IRL to the incremental setting, and with an unknown observation model. Our method engages in robust incremental IRL with a maximum entropy estimation of the observation model (labeled as RIMEO), and is an instantiation of the recent I2RL framework [5]. We formulate it as a sequence of incremental learning sessions where each session is a two-step process – maximum entropy learning of the observation model of the learner, followed by inversely learning the reward function of the expert by utilizing the learned observation model.

Using noisy observations of a human engaged in sorting onions, we evaluate the performance of RIMEO in a Gazebo sim of the onion-sorting domain in which Sawyer processes the observations incrementally and eventually imitates the task of sorting the onions. We show that it generates a significantly improved sorting performance compared to a recent end-to-end IRL baseline that accepts raw images as input. As this and other IRL methods do not directly learn an observation model, we also evaluate the comparative advantage of RIMEO against a custom baseline which learns a less sophisticated observation model. Similar results are observed in a second task of a Turtlebot learning to breach a patrol by another. Furthermore, we formally show that RIMEO's data likelihood improves monotonically with more sessions, and it admits a probabilistic bound for sample complexity.

## 2 Related Work

Observation noise is not widely addressed by the IRL literature, which tends to assume that the state-action data is accurate. However, this assumption may not hold in robotic applications where the learner is using (noisy) sensors to watch the expert (a paradigm called learn-from-observation (LfO)). Robust IRL [4] was an early technique, which estimated the ground truth underlying the noisy observations using a known observation model. But, it did not address cases where the observation model is unknown, or when the data is only available incrementally. More recently, a hierarchical Bayesian model [6] was presented to enable IRL in partially-controlled environments where elements that confound the observations of the expert may be present. The model offers a way to sample the source of each observation and a hyperprior over the observation models is defined. Bayesian IRL has also recently been extended to contexts involving both occluded and noisy observations using marginal maximum-a-posteriori inference [7]. However, the observation or sensor model is assumed to be known in order to utilize this method.

We also note IRL methods that learn the stochastic transition dynamics, either by using importance sampling [8] or by decomposing each transition into its component features and learning from feature aggregates [9]. Adapting the former method for unknown observation models requires a preexisting near-perfect base observation model, which may get unrealistic. We take inspiration from

the latter method, where, analogously to Bogert and Doshi [9], we decompose an observation into its component observation features followed by learning the feature probabilities, and the learner generalizes from that information.

## 3  Background

Inverse RL aims to find a reward function under which the observed behavior of the expert is optimal, with the behavior modeled as an incomplete Markov decision process (MDP) $\langle S, A, T \rangle$ [10, 2]. Abeel and Ng [11] modeled the reward function as a linear combination of $K$ binary features, $\phi_k$, each of which maps a state and an action to a value in $\{0,1\}$. The linearized reward function is $R_{\boldsymbol{\theta}}(s, a) = \boldsymbol{\theta}^T \phi(s, a) = \sum_{k=1}^{K} \theta_k \cdot \phi_k(s, a)$, where $\theta_k$ are the *feature weights* in $\boldsymbol{\theta}$. The learner's task is to find a $\boldsymbol{\theta}$ that makes the observed behavior optimal in the completed MDP $\langle S, A, R_{\boldsymbol{\theta}}, T \rangle$.

Let $\Xi$ be the space of all possible ground truth trajectories of finite time-step length $T$ (i.e., sequence of $T$ $\langle s, a \rangle$ pairs) executable by the expert, and $\Xi_d$ be the set of trajectories observed by the learner. The cumulative value of any feature $\phi_k$ for trajectory $\xi$ is given by a function $f_k : \Xi \to \mathbb{R}$ defined as $f_k(\xi) = \sum_{\langle s,a \rangle_t \in \xi} \gamma^t \phi_k(\langle s, a \rangle_t)$, where $t$ is the temporal index of a state-action pair in the trajectory. Ziebart et al. [12] solves IRL by finding the probability distribution over $\Xi$ that exhibits the maximum entropy and is constrained to match the observed feature expectations:

$$\max_{\Delta} \left( -\sum_{\xi \in \Xi} P(\xi) \log P(\xi) \right) \textbf{ subject to } \sum_{\xi \in \Xi} P(\xi) = 1, \textbf{ and } E_{\Xi}[\phi_k] = \hat{\phi}_k \forall k = 1 \ldots K \quad (1)$$

Here, $\Delta$ is the space of all distributions over the finite set $\Xi$, and $E_{\Xi}[\phi_k] \triangleq \sum_{\xi \in \Xi} P(\xi) f_k(\xi)$. The learner empirically estimates the true feature expectation as $\hat{\phi}_k = \frac{1}{|\Xi_d|} \sum_{\xi \in \Xi_d} f_k(\xi)$.

### 3.1  Robust IRL with a Known Observation Model

We define the observation model as a function that gives the likelihood of a perceived state-action pair $\langle s, a \rangle_o$ given the demonstrated (ground truth) state-action pair $\langle s, a \rangle_g$ [1]. That is,

$$O_b(\langle s, a \rangle_o, \langle s, a \rangle_g) \triangleq P(\langle s, a \rangle_o \mid \langle s, a \rangle_g) \quad (2)$$

the RHS being the probability of the learner observing $\langle s, a \rangle_o$ when the expert actually demonstrated $\langle s, a \rangle_g$. Extending this observation model to trajectories, for a ground truth trajectory $\xi_g \in \Xi$, the probability of observing trajectory $\xi_o \in \Xi_d$ is

$$P(\xi_o | \xi_g) = \prod_{\langle s,a \rangle_o \in \xi_o, \langle s,a \rangle_g \in \xi_g} P(\langle s, a \rangle_o \mid \langle s, a \rangle_g). \quad (3)$$

The batch learning method of Robust IRL [4] uses this observation model to learn the reward function under noisy observations. In the process, it estimates the $k^{th}$ feature expectation as

$$\hat{\phi}_{\boldsymbol{\theta},k} = \frac{1}{|\Xi_d|} \sum_{\xi_o \in \Xi_d} \sum_{\xi_g \in \Xi} P(\xi_g | \xi_o; \boldsymbol{\theta}) f_k(\xi_g) \quad (4)$$

where $P(\xi_g | \xi_o; \boldsymbol{\theta}) = \eta P(\xi_o | \xi_g) P(\xi_g; \boldsymbol{\theta})$ and $P(\xi_g; \boldsymbol{\theta})$ is the probability of the generation of a trajectory $\xi_g$ using a policy computed with $\boldsymbol{\theta}$. Then the program of (1) becomes

$$\max_{\Delta} \left( -\sum_{\xi_o \in \Xi_d, \xi_g \in \Xi} P(\xi_o, \xi_g) \log P(\xi_o, \xi_g) \right) \textbf{ subject to } \sum_{\xi_o \in \Xi_d, \xi_g \in \Xi} P(\xi_o, \xi_g) = 1 \textbf{ and } E_{\Xi}[\phi_k] = \hat{\phi}_{\boldsymbol{\theta},k} \ \forall k \quad (5)$$

where $E_{\Xi}[\phi_k] = \sum_{\xi_o \in \Xi_d} \sum_{\xi_g \in \Xi} P(\xi_o, \xi_g) f_k(\xi_g)$. We extend this optimization problem to the case of an unknown observation model and online learning in Sections 4 and 5, respectively.

### 3.2  I2RL Framework

Arora et al. [5] formulate incremental IRL as a sequence of sessions with $R_i$ as the $i^{th}$ session's estimate of the expert's reward function. A formal definition of incremental IRL follows.

---

[1] Although we use a state-action pair as an observation, the robust-IRL method is generalizable to any form of observations [4].

DEFN 3.1 (SESSION). *A session of* I2RL, $\zeta_i(MDP_{/R}, \Xi_{d,i}, \alpha_{1:i-1}, R_{i-1})$, $i \in \mathbb{N}^+$, *takes as input the expert's incomplete MDP, the current ($i^{th}$) demonstration, $\Xi_{d,i}$, the sufficient statistic summarizing previous $i-1$ sessions ($\alpha_{1:i-1}$), and the reward function estimated previously. It yields a revised estimate of the expert's reward function, $R_i$.*

DEFN 3.2 (I2RL). *Incremental IRL is a sequence of learning sessions $\{\zeta_1(MDP_{/R}, \Xi_{d,1}, \emptyset, R_0), \zeta_2(MDP_{/R}, \Xi_{d,2}, \alpha_{1:1}, R_1), \zeta_3 \ldots, \}$, which continue indefinitely or until a stopping criterion assessing convergence is met.*

As a measure of how well the learned reward function explains the observed data, we may use the log likelihood of the accumulated demonstration received up to the $i^{th}$ session.

## 4    Learning the Observation Model

Our overall approach is to approximate the unknown observation model using an underlying feature set shared among observations, and then fit it to the noisy observations via maximum entropy optimization. Let $\Psi$ be this set of $m$ observation features. Each feature, $\psi_j \in \Psi$, $j \in \{1, 2, \ldots m\}$ is a predicate that is true if and only if a specific characteristic represented by that feature appears in the observed state-action pair under consideration. Let $\psi^{o,g}$ be the set of indicators, whose member $\psi_j^{o,g}$ assumes a value based on comparing the value of a feature on ground truth pair $\langle s, a \rangle_g$ and on observation $\langle s, a \rangle_o$: $\psi_j^{o,g} = \mathbb{1}_{\psi_j(\langle s,a \rangle_o) = \psi_j(\langle s,a \rangle_g)}$, where $\psi_j \in \Psi$ and $j \in \{1, 2, \ldots, m\}$.

We assume that the learner knows the feature set and that the features are independent. For each $\langle s, a \rangle_o \in \Xi_d$, $\langle s, a \rangle_g \in \Xi$, we approximate the observation probability (Eq. 2) to be the product of (unknown) probabilities of the features $\psi_j$ according to the values taken by the corresponding indicators $\psi_j^{o,g}$,

$$O_b(\langle s, a \rangle_o, \langle s, a \rangle_g) \approx \prod_{\psi_j \in \Psi, s.t. \psi_j^{o,g} = 1} P(\psi_j) \cdot \prod_{\psi_j \in \Psi, s.t. \psi_j^{o,g} = 0} P(\bar{\psi}_j) \tag{6}$$

Here $P(\psi_j)$ can be interpreted as the the probability of the $j$th feature being non-noisy, and $P(\bar{\psi}_j) = 1 - P(\psi_j)$. Though the learner knows the features, it cannot see the noise-free ground truth state-action pairs and those pairs not included in the demonstration. Therefore, obtaining a *complete* observation model requires inferring a distribution over observation features for all state-action pairs, regardless of whether they are observed or not.

**Maximum-Entropy Observation Model**    If deep neural network based models such as YOLO [13] and SA-Net [14] are used to process raw videos of expert performing the task, we may obtain stochastic confidence scores for the state-action labels which we denote using the function $c(\langle s, a \rangle)$ (Fig. 2). The likelihood of an observation $\langle s, a \rangle_o$ given any ground truth $\langle s, a \rangle_g$ can be empirically estimated as a running average of the confidence scores from observed trajectories:

$$\hat{O}(\langle s, a \rangle_o, \cdot) \triangleq \frac{\sum_{\xi_o \in \Xi_d} \sum_{\langle s,a \rangle \in \xi_o} c(\langle s, a \rangle) \mathbb{1}_{\langle s,a \rangle_o = \langle s,a \rangle}}{\sum_{\xi_o \in \Xi_d} \sum_{\langle s,a \rangle \in \xi_o} \mathbb{1}_{\langle s,a \rangle_o = \langle s,a \rangle}}. \tag{7}$$

Utilizing the principle of maximum entropy (a form of least commitment), the learner infers a unique distribution over the observation features $\Psi$ which has the maximum entropy, while satisfying the constraint imposed by Eqs. 6 and 7: that the aggregated observation feature probabilities match the corresponding empirical estimates of observation likelihoods. As the ground truth is not available to the learner, it may instead consider possible state-action pairs that could have been in the input, $\langle s, a \rangle_{\hat{g}}$ where $\hat{g}$ is the estimated ground truth from the expert, rather than considering all possible state-action pairs. The optimization problem is:

$$\max_{\Delta_\psi} - \sum_{\psi_j \in \Psi} \left( P(\psi_j) \log P(\psi_j) + P(\bar{\psi}_j) \log P(\bar{\psi}_j) \right) \textbf{ subject to}$$

$$\prod_{\psi_j^{o,\hat{g}} = 1} P(\psi_j) \prod_{\psi_j^{o,\hat{g}} = 0} P(\bar{\psi}_j) = \hat{O}(\langle s, a \rangle_o, \cdot), \;\; \forall \langle s, a \rangle_o \in \Xi_d \textbf{ and } P(\psi_j) + P(\bar{\psi}_j) = 1 \;\; \forall \psi_j \in \Psi.$$

The Lagrangian relaxation of the above program is:

$$\mathcal{L}(\Psi, \mathbf{v}, \boldsymbol{\lambda}) = - \left( \sum\nolimits_{\psi_j \in \Psi} P(\psi_j) \log P(\psi_j) + P(\bar{\psi}_j) \log P(\bar{\psi}_j) \right)$$

$$+ \sum_{\langle s,a \rangle_{\hat{g}} \in \Xi, \langle s,a \rangle_o \in \Xi_d} v^{o,\hat{g}} \left( \prod_{\psi_j^{o,\hat{g}}=1} P(\psi_j) \prod_{\psi_j^{o,\hat{g}}=0} P(\bar{\psi}_j) - \hat{O} \right) + \sum_{\psi_j \in \Psi} \lambda_{\psi_j} \left( \left( P(\psi_j) + P(\bar{\psi}_j) \right) - 1 \right) \tag{8}$$

where $\mathbf{v}, \boldsymbol{\lambda}$ are vectors of multipliers $v^{o,\hat{g}}$ and $\lambda_{\psi_j}$.

Notice that the same feature $\psi_j$ could be shared by multiple, distinct $\langle s, a \rangle_{\hat{g}}$ and observation $\langle s, a \rangle_o$ pairs, which then activate $\psi_j^{o,\hat{g}}$. Several of these active indicators for various $j$ can be present in the LHS of the first constraint and these receive a probability in aggregate from $\hat{O}$. Bard [15] was the first to note that a maximum entropy approach is beneficial when just the aggregate probabilities are available. The critical points of a Lagrangian often occur at saddle points rather than at local maxima or minima [16], and numerical optimization solvers may not find the saddle points. Therefore, we modify the Lagrangian objective to make critical points occur at local optima, $\mathcal{L}' = \sqrt{ \left( \frac{\partial \mathcal{L}}{\partial P(\psi_j)} \right)^2 + \left( \frac{\partial \mathcal{L}}{\partial v^{o,\hat{g}}} \right)^2 + \left( \frac{\partial \mathcal{L}}{\partial \lambda_{\psi_j}} \right)^2 }$, and use limited memory Broyden-Fletcher-Goldfarb-Shanno (L-BFGS) [17, 18] unconstrained optimization solver to learn the maximum-entropy distribution $P^*$. Equation 6 then obtains the observation model from this distribution, which paves the way to obtain $P(\xi_o | \xi_g)$ using Eq. 3 making it possible to apply the previous Robust IRL technique.

## 5 Robust Incremental IRL

To extend this process to an online setting, the learner uses $P_i^*$ learned at the beginning of session $i$ to estimate the feature expectations:

$$\hat{\phi}_{\boldsymbol{\theta}^i, k}^i \triangleq \frac{1}{|\Xi_{d,i}|} \sum_{\xi_o \in \Xi_{d,i}} \sum_{\xi_g \in \Xi} P_i^*(\xi_g | \xi_o; \boldsymbol{\theta}) f_k(\xi_g) = \frac{1}{|\Xi_{d,i}|} \sum_{\xi_o \in \Xi_{d,i}} \sum_{\xi_g \in \Xi} \eta P_i^*(\xi_o | \xi_g) P(\xi_g; \boldsymbol{\theta}) f_k(\xi_g). \tag{9}$$

Let $\Xi_{d,1:i}$ be the cumulative demonstration up to the $i^{th}$ session, and $\hat{\phi}_{\boldsymbol{\theta}^i, k}^{1:i}$ be the corresponding cumulative feature expectations. The learner updates the latter from session to session as

$$\hat{\phi}_{\boldsymbol{\theta}^i, k}^{1:i} \triangleq \frac{1}{|\Xi_{d,1:i}|} \sum_{\xi_o \in \Xi_{d,1:i}} \sum_{\xi_g \in \Xi} P_{1:i}^*(\xi_g | \xi_o; \boldsymbol{\theta}) f_k(\xi_g) = \frac{1}{|\Xi_{d,1:i}|} \left( \sum_{\xi_o \in \Xi_{d,1:i-1}} \sum_{\xi_g \in \Xi} P_{1:i-1}^*(\xi_g | \xi_o; \boldsymbol{\theta}) f_k(\xi_g) \right.$$

$$+ \sum_{\xi_o \in \Xi_{d,i}} \sum_{\xi_g \in \Xi} P_i^*(\xi_g | \xi_o; \boldsymbol{\theta}) f_k(\xi_g) \right) = \frac{1}{|\Xi_{d,1:i-1}| + |\Xi_{d,i}|} \left( |\Xi_{d,1:i-1}| \, \hat{\phi}_{\boldsymbol{\theta}^{i-1}, k}^{1:i-1} + |\Xi_{d,i}| \, \hat{\phi}_{\boldsymbol{\theta}^i, k}^i \right). \tag{10}$$

### 5.1 The RIMEO Method

Following Def. 3.1, a session of RIMEO is $\zeta_i(MDP_{/R}, \Xi_{d,i}, \alpha_{1:i-1}, \boldsymbol{\theta}^{i-1})$ where sufficient statistic $\alpha_{1:i-1}$ contains the summary up to the last session $(i-1)$, viz., cumulative size of the demonstration set $(|\Xi_{d,1:i-1}|)$, the learned $P^*$ values from the optimization in Eq. 8, and the cumulative feature expectations $(\hat{\phi}_{\boldsymbol{\theta}^{i-1}}^{1:i-1})$. Specifically, $\alpha_{1:i-1} = (|\Xi_{d,1:i-1}|, P_{1:i-1}^*, \hat{\phi}_{\boldsymbol{\theta}^{i-1}}^{1:i-1})$. This information, together with $\Xi_{d,i}$ and $\boldsymbol{\theta}^{i-1}$, is sufficient for Eq. 9 and thence Eq. 10. For demonstration $\Xi_{d,i}$ of the current session $i$, the scores $c(\langle s, a \rangle)$ for input $(s, a)$ pairs are available from the perception pipeline. The process starts by computing $\hat{O}$ using the scores and Eq. 7. The resulting $\hat{O}$ is plugged into Eq. 8 to get $\nabla \mathcal{L}'$, using which $P_{1:i}^*$ is updated by the L-BFGS gradient ascent starting with the initial value of $P_{1:i-1}^*(\psi)$. The learned $P_{1:i}^*$ are then used to compute the updated observation model probabilities $P(\langle s, a \rangle_o | \langle s, a \rangle_g)$ (Eqs. 2, 6). This extends to a corresponding update for full trajectories, $P_{1:i}(\xi_o | \xi_g, \boldsymbol{\theta})$ (Eq. 3). Solving Eq. 9, the learner uses MCMC to sample ground truth trajectories from $P_{1:i}^*(\xi_g | \xi_o, \boldsymbol{\theta})$ for computing the feature expectations $\hat{\phi}_{\boldsymbol{\theta}^i}^i$, followed by computation of $\hat{\phi}_{\boldsymbol{\theta}^i}^{1:i}$ via incremental update (Eq. 10). Finally, the latter is used for the main Robust IRL optimization (Eq. 5) via L-BFGS gradient descent to learn

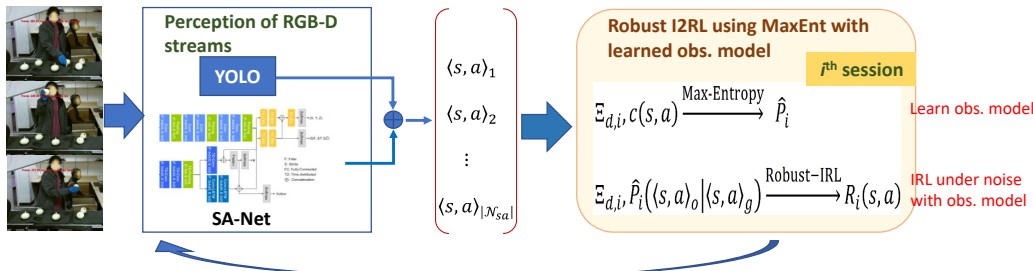

Figure 2: An overview of RIMEO. On the left is an image stream depicting the hidden ground truth, $\langle s, a \rangle_g$, of the expert's behavior. As the perception process is not perfect, the input to the learning algorithm is noisy. This input is an array of possibly incorrect observations and corresponding (classification) scores, $c(\langle s, a \rangle_o)$, that measure possible match with unknown $\langle s, a \rangle_g$.

the reward weights ($\boldsymbol{\theta}^i$) and corresponding policy $\pi_i$. Due to space constraint, we present the detailed algorithm with links to our GitHub code in Appendix A included in the supplementary file which is available at `http://thinc.cs.uga.edu/files/abdCoRL22-supp.pdf`.

## 5.2 Convergence Properties of RIMEO

While some of the results in this section are analogous to those for I2RL [5], Lemma 2 is the main original result of this paper since it incorporates the additional uncertainty due to noisy observations and an unknown observation model. Due to space constraint, we present all proofs in the Appendix B. The demonstration log-likelihood for session $i$ is given by $LL(\boldsymbol{\theta}^i | \Xi_{d,i}, \alpha_{1:i-1}, \boldsymbol{\theta}^{i-1})$.

LEMMA 1 (MONOTONICITY). *The demonstration likelihood increases monotonically with each new session, $LL(\boldsymbol{\theta}^i | \Xi_{d,i}, \alpha_{1:i-1}, \boldsymbol{\theta}^{i-1}) - LL(\boldsymbol{\theta}^{i-1} | \Xi_{d,i-1}, \alpha_{1:i-2}, \boldsymbol{\theta}^{i-2}) \geqslant 0$, when $|\Xi_{d,1:i-1}| \gg |\Xi_{d,i}|$.*

Note that feature expectations estimated under noisy perception with an estimated observation model, $\hat{\phi}^{1:i}_{\boldsymbol{\theta}^i,k}$ from Eq. 10, is an approximation of the expectations when the observation model is known accurately. $\hat{\phi}^{1:i}_{\boldsymbol{\theta}^i,k}$ is computed by sampling the hidden ground truth $\xi_g$ from $P(\xi_g | \xi_o, \boldsymbol{\theta}^{i-1})$. This, in turn, is an approximation of the expectations estimated under noise-free perception, $\hat{\phi}^{1:i}_k$. As in I2RL [5], we assume that the latter estimation using $n_s$ samples obeys Hoeffding bounds, i.e., $P(|\hat{\phi}^{1:i}_k - \hat{\phi}^{1:i}_{\boldsymbol{\theta}^i,k}| \leq \varepsilon_s) \geq 1 - \delta_s$, where $\delta_s = 2K \exp(-2n_s \varepsilon_s^2)$. Additionally for this paper, we assume that our observation model estimation with $n_o$ samples also obeys Hoeffding bounds as $P(\max_{\langle s,a \rangle_o} |O - \hat{O}| \leq \varepsilon_o) \geq 1 - \delta_o/K$, where $O$ is the true observation model (of which $\hat{O}$ is an estimate), and $\delta_o = 2K|\Psi| \exp(-2n_o \varepsilon_o^2)$. We also assume that all features in $\Psi$ are observed in the very first session. Then the following lemma holds.

LEMMA 2 (CONSTRAINT BOUND ). *Under the assumptions stated above, with probability at least $\max(0, 1 - \delta_r)$: $\left| (1 - \gamma)(E_\Xi[\phi_k] - \hat{\phi}^{1:i}_{\boldsymbol{\theta}^i,k}) \right|_1 \leqslant \varepsilon_r, \forall k \in \{1, 2 \ldots K\}$, where $\delta_r = \delta + \delta_s + \delta_o$, $\varepsilon_r = \varepsilon + \varepsilon_s + L|\Psi|\varepsilon_o$, $L$ is the longest trajectory length, and $\varepsilon, \delta$ are as defined in Theorem 1 in [5].*

Lemma 2 allows us to probabilistically bound the error in the log likelihood for RIMEO, leading to the main result below specifying the confidence in its convergence.

THEOREM 1 (CONFIDENCE ). *Let $\varepsilon_r, \delta_r$ be as defined in Lemma 2, and $\boldsymbol{\theta}^i$ be the solution of session $i$ for RIMEO. Then $LL(\boldsymbol{\theta}_E | \Xi_{d,1:i}) - LL(\boldsymbol{\theta}^i | \Xi_{d,i}, \alpha_{1:i-1}, \boldsymbol{\theta}^{i-1}) \leq \frac{2K\varepsilon_r}{(1-\gamma)}$, with confidence at least $\max(0, 1 - \delta_r)$, where $\boldsymbol{\theta}_E$ are the true weights of the expert.*

## 6 Experiments

Our primary domain for evaluating RIMEO is the onion sorting illustrated in Fig 1. The expert aims to identify blemished onions from a collection of onions on a table, and drop blemished ones in a bin while allowing others to pass. A state of the task is composed of these features: *onion location* and *end-effector location* {on table, picked up in hover region, at eye level (under inspection), or near the

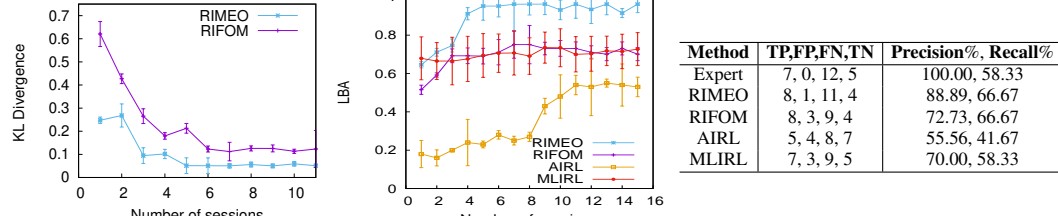

Figure 3: (**left**) Divergence between $P(\psi)$ and $\hat{P}(\psi)$ for simulated onion sorting data. (**middle**) LBA of the learned policy on the human demonstration data. (**right**) Column label TP denotes true positive (# blemished onions in bin), FP is false positive (# good onions in bin), TN true negative (# good onions remaining on table), and FN denotes false negatives (# blemished onions remaining on table).

bin for blemished onion}, and *quality prediction* {blemished, unblemished, or unknown}, resulting in 22 states. The actions involve attending to a new onion on the table, picking it up, bringing the grasped onion closer to eye and inspecting it, placing it in the bin, and placing it back on the table. We utilize $K = 11$ predicates as reward features, $\phi_k(s, a)$, listed in Appendix C. To learn the observation model, we use 8 binary predicates $\psi_j$ as observation features, [2] also listed in Appendix C.

**Baselines:** We use MLIRL [19] and AIRL [20] as baselines to compare with RIMEO. AIRL is state-of-the-art and incremental because the learned weights of the discriminator and generator nets are a sufficient statistic. To our knowledge, no other IRL method infers an observation model. Therefore, we create a custom baseline using the frequentist approach, which uses robust I2RL with a frequentist estimation of the observation model (RIFOM). While RIMEO solves an optimization problem to learn $P_i^*$, RIFOM assumes that the ground truth is the same as the observation and learns $P_i^*$ as the running average of normalized frequencies of observation features activated in the demonstration.

**Setup:** We use videos of a human sorting onions (Fig. 1) as the training demonstration. We utilize SA-Net [14] and YOLO [13] for identifying the sequences of states in the videos. The perception pipeline considers the state with the highest prediction score $c(s, a)$ as the current state in the input trajectory $\xi_o \in \Xi_d$. This noisy trajectory and the confidence scores are input to I2RL sessions of all methods. The domain is simulated in Gazebo 7 (ROS Melodic) with the cobot Sawyer as the learner. The experiments were run on a desktop with Intel Xeon CPU (E5-1603, 2.8GHz) and 16GB RAM.

**Performance Results:** We evaluate the divergence of the learned $P^*(\psi)$ from a *simulated* dataset for which the true distribution $P(\psi)$ is known. We introduce noise as $P(\psi_j(\langle s, a\rangle_o) \neq \psi_j(\langle s, a\rangle_g) = 0.3$ for $j = BlemishedOnion, OnionToBin, OnionToTable$, and 0 for other features. Figure 3(left) shows the KL divergences between the distributions $P(\psi)$ and $P_i^*(\psi)$ with sessions $i$. Each data point is averaged over 50 runs, and each session has 2 trajectories of size 3 as input. The divergence for RIMEO is significantly lower than that of RIFOM for all the sessions, indicating that RIMEO learns a more accurate $P_i^*$. Observe that the divergence starts stabilizing after the sixth session. As AIRL or MLIRL do not directly learn an observation model, we do not report their divergence measures.

Next, we measure the *learned behavior accuracy* (LBA), which is the proportion of states in which the action prescribed by the learned policy matches the actual action of the expert. Figure 3(middle) shows the average LBA for our *human demonstration data set* as the number of sessions increases, for all three methods. While AIRL's LBA improves initially (until session 11), it flattens out to a value (maximum 0.48) that is considerably lower than the other two methods. On average, AIRL is also slower to converge (about 8 minutes) as compared to RIMEO (96.69 seconds). This performance is likely due to the need of neural network based adversarial techniques for more training data and that AIRL suffers when the training data (that is used by the discriminator) is noisy. Furthermore, RIFOM's LBA flattens out around 23% lower than RIMEO's. This is because the maximum-entropy optimization of the underlying observation features in RIMEO generalizes the learned information across $(s, a)$ pairs not present in the demonstration, but the frequentist baseline fails to generalize. RIMEO eventually reaches an LBA of 96% whereas RIFOM's stays around 70%. However, as RIMEO solves an additional optimization problem for learning $\hat{P}$, it takes longer to converge (96.69 seconds, on average) compared to RIFOM's (85.99 seconds). MLIRL's LBA is similar to RIFOM's,

---

[2]We used Pearson product-moment correlation (0.03 with p-value 0.04, averaged over all pairs of features from demonstration) to confirm that these observation features are sufficiently independent.

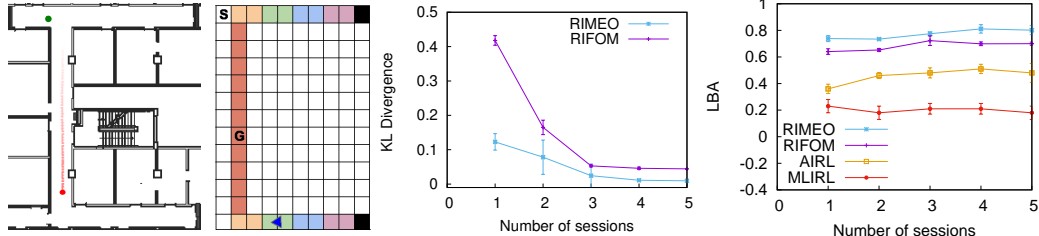

Figure 4: (**left**) Patroller (red) navigates hallways [21], and learner (green) aims to pass undetected. The 5 colored regions (long hallway, turning points, and three small divisions in small hallways on both sides) define movement and turn-around features. S and G are start and goal locations for the learner. (**middle**) Divergence between $P(\psi)$ and $\hat{P}(\psi)$, and (**right**) LBA of the learned policies on Gazebo simulation data of learner.

but its high variance and inability to improve its performance can be attributed to observation noise. As it is not incremental (it was run with accumulating data in batch mode), runtime comparison is not meaningful and hence not reported.

To evaluate sorting performance, Sawyer uses the learned policies to sort a fixed number of onions. For object tracking and onion classification, we use a trained YOLO model with a simulated camera in Gazebo. For all techniques, we execute the policy learned in the final session, to sort 6 moving collections of 4 onions each (2 blemished and 2 good). The table in Fig. 3(right) shows the average precision and recall of the expert and the 4 learning methods. As the expert aims to inspect every onion whose blemish is not immediately visible, which is slow, it exhibits high precision but low recall. RIMEO's precision is closest to the expert's due to lower false positives. AIRL performs worse in both precision and recall having higher false positives and true negatives. This is consistent with its low LBA, indicating its reduced ability to successfully sort onions under noise. Among the two robust I2RL methods, as the steps after learning the observation model are the same, we attribute the better precision of RIMEO to a more accurate observation model.

**Perimeter Patrol Domain:** To establish the generality of RIMEO, we evaluate it in a second domain: a simplified version of the perimeter patrol domain [21]. It involves two Turtlebots (in Gazebo), one patroller continuously navigates a hallway in cyclic trajectories; the other is a learner tasked with reaching a goal location without being spotted by the patroller (Fig. 4). The learner must solve its own distinct decision-making problem (modeled as another MDP), which is dependent on correctly predicting the patrol. The latter can be estimated from inferring the patroller's preferences given that the learner knows its dynamics. The state of patroller in the MDP has three dimensions $\langle x, y, \theta \rangle$, which gives the $x$ and $y$ coordinates of the cell decomposition of the corridors and hallways, and $\theta$ is the orientation of the patroller. The domain has 124 states. The patroller can execute one of four possible actions: move-forward, turn left or turn right 90 degrees, and stop. The reward function utilizes 6 binary state-action feature functions, described in Appendix D. The observation feature set $\Psi$ contains 4 binary predicates also described in Appendix D.

We introduce noise as $P(\psi_j(\langle s, a \rangle_o) \neq \psi_j(\langle s, a \rangle_g) = 0.3$ for $j = $ *MoveForward, TurnRight*, and 0 for other features. The two figures (right) in Fig. 4 show the performance of both robust I2RL methods in perimeter patrol. As in onion-sorting, KL divergence and LBA for RIMEO is better than RIFOM, AIRL and MLIRL. These results in two unrelated tasks indicate that the presented method is sufficiently general for broad applicability.

## 7 Conclusion

We introduced an online IRL method that accommodates noisy observations and infers the unknown observation model by maximum entropy optimization. Not only does this method fill a gap in the prior work on online IRL but it also makes online learning more pragmatic, as evaluated in two robotics tasks of post-harvest onion sorting and breaching a patrol.

**Limitations:** The current set up restricts the observation features to binary functions. The assumption of feature independence and the key approximation of Eq. 6 may not be easily satisfied, and could have contributed to RIMEO's learned observation model not being accurate (KLD=0). To meet the assumption that all $\psi$ are observed in the first session, it may need to contain more demonstrations.

**Acknowledgments**

We thank the anonymous reviewers for helpful comments and suggestions. This research was supported in part by a grant from NSF #IIS-1830421 and a Phase 1 grant from the GA Research Alliance to PD.

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
