# OpenReview forum: "Online Inverse Reinforcement Learning with Learned Observation Model"
_robot-learning.org/CoRL/2022/Conference — CoRL 2022 Poster_

### Official Review · Reviewer_MqYS · 2022-07-29

**Originality:** Fair
**Technical Quality:** Good
**Clarity Of Presentation:** Fair
**Impact:** 3

**Recommendation:**

Weak Reject: I recommend rejecting the paper, but will not argue for my recommendation if the majority of other reviewers have a different opinion.

**Summary:**

This paper introduces an inverse reinforcement learning method, which learns an observation model, to deal with real world robotics applications with noisy observations. The observation model is learned using the maximum entropy principle and is then used to learn a reward function. The experiments are conducted on two tasks: (1) post-harvest onion sorting using human demonstrations; (2) a perimeter patrol task. On the presented tasks, this approach outperforms a state-of-the-art adversarial inverse reinforcement learning method (AIRL) and a custom baseline, which also uses an observation model (yet a less sophisticated one).


**Issues:**

**Issues**:
- the notation is very confusing. The combination of $\phi$ for the binary features, $\psi_j$ for the binary predicates and $\psi_J^{o,g}$ is not very straight forward to understand. In addition to that, Figure 2
confuses even more, where $\psi^{1,g}_j$ (where does the 1 come from?) is introduced, of which I am not sure of what it means. It would help a lot to create a more informative Figure presenting the general approach
a bit more clearly. Figure 2 is not well made in my opinion.
- Equation (6) uses the binary predicates $\psi_j (<s, a>)$ to define the indicators. Why is a different notation used for indicators in equation (8)s?
- Use a consistent math font, even in the Figures.
- It needs to be explained in greater detail, how the confidence intervals are calculated and the networks are trained.
- Implementation details, also for AIRL, on the experiments are missing. A judgment on AIRL's computational performance without giving implementation details is not very helpful.
- I don't think that AIRL is a good baseline for this task, due to the small state and action states. I would like to see a comparison to an approach made for discrete state and action spaces.


**Minor Issues**:
- please blur faces in Figures next time to ensure double blindness (even if the pictured persons are not the authors, but lab members etc.)

**Quality Of The Limitations Section:**

Additional details required

**Reviewer Expertise:**

3: The reviewer is fairly confident that the evaluation is correct

**Robotics Focus:**

Relevant but unlikely to deploy to hardware in near future

**Strengths And Weaknesses:**

**Strengths**:
- the task of inferring a reward under noisy expert demonstration is of general interest

**Weaknesses**:
- the paper is not very well written. The paper is overly complex with a confusing set of notations.
- the contribution is minor, only the observation model is introduced to the online inverse reinforcement learning setting.
- the application scope of the presented algorithms is very limited --> discrete states and actions, with binary observation features.
- the presented robotic tasks is solved on a very high-level.
- the comparisons made to state-of-the-art are made on very specific tasks, with very specific features. It is hard to judge on the general performance of this approach.
- no sufficient implementation details are given (e.g., in the appendix).
- no video and no code is provided.

**Summary Of Recommendation:**

From my perspective, the general approach of inferring a reward under noisy expert demonstration is interesting and promising. However, the paper introduced a very limited approach and benchmarked on very specific task with a small number of highly engineered features and observations. This makes it very hard to judge on the performance of the presented approach in comparison to the state-of-the-art.
Moreover, the paper is unnecessarily complex written making it hard to follow and not straightforward to understand. The notation needs to be revised from my perspective.

---

> ### Author Response · Authors · 2022-08-27
> **Authors' Response**
>
> As far as we can tell, our notations are standard, only extended to accommodate the additional notion of sessions. Notation heaviness in some places allowed us to be less verbose and meet the page limitation. Following another reviewer’s suggestion, we plan to move the exhaustive feature lists to the Appendix, to buy space to reduce notational clutter, and for a less formula-heavy exposition.
>
> While our algorithm requires discrete states/actions, the problems it can solve are not restricted to discrete states/actions, thanks to SA-Net as an intermediary. As shown in Fig. 2, we can process raw RGB-D streams (of demonstrations), which is continuous data. We believe it is a strength to solve complex AI problems for applications in robotics at a higher level of abstraction, since solutions are unscalable otherwise.
>
> We have shown results in two different, unrelated tasks, to demonstrate that our method is sufficiently general for broader applicability. We have also offered some ways to construct features in other tasks, in response to another reviewer.
>
> As explained in the Appendix (originally submitted with the paper), we intend to publicly release our entire code and dataset should the paper be accepted. This is why we did not spend time/space on describing implementation details. Our code is already on GitHub, split into multiple repos, just not anonymized.
>
> In the paper’s Fig. 2, $\psi_j^{1,g}$ is $\psi_j^{o,g}$ for observation $o=1$. This corresponds to the perception process output $\langle s,a\rangle_1$. Therefore, for $\langle s,a\rangle_2$ there would be $\psi_j^{2,g}$, and so on. This is now moot as Fig. 2 has been redone; please see the attachment.
>
> We fail to see any difference/inconsistency in the indicator notation between eqns. 6 and 8. They are applied to different predicates in those two eqns. There is a difference in the paper’s Fig. 2, but it has now been redone.
>
> All confidence intervals are 1 standard deviation from the mean, based on 50 independent trials. We are not sure which network’s training details are being sought. For any external methods we cite, we use the corresponding authors’ original code (from GitHub, personal correspondence, etc.) with mostly default parameters. This is true for SA-Net, AIRL (mostly; see deviations below), and MLIRL. Our methods, RI2RL-MEOM/FQOM don’t use neural networks. For AIRL, we used the implementation from https://github.com/HumanCompatibleAI/imitation, and changed the expert’s batch size (number of expert samples per discriminator update) from 1024 to 4 (must be an integer power of 2) to closely match the trajectory lengths in our demonstrations.
>
> We are now adding another baseline suitable for discrete state and action spaces – MLIRL [VRoman, M. C. 2014. Maximum Likelihood Inverse Reinforcement Learning. Ph.D. Dissertation, Rutgers University] – as you suggested. So far we have obtained MLIRL comparison for just the perimeter patrol domain (please see the attachment); experiments in the  onion sorting domain are on-going. Before trying MLIRL, we tried a more recent algorithm – Scalable BIRL [ICLR 2021; https://github.com/XanderJC/scalable-birl] – but failed to run it due to JAX compatibility issues.

---

### Official Review · Reviewer_1CPf · 2022-07-29

**Originality:** Good
**Technical Quality:** Fair
**Clarity Of Presentation:** Fair
**Impact:** 3

**Recommendation:**

Weak Reject: I recommend rejecting the paper, but will not argue for my recommendation if the majority of other reviewers have a different opinion.

**Summary:**

This work deals with inverse reinforcement learning (IRL), specifically in robotic manipulation tasks where the observation model is noisy.  They state this is a more realistic scenario since real-world robotic perception is noisy and models therefore must be updated incrementally.  Previous works in IRL assume that the observation model is known, and they extend this work to additionally learning an unknown observation model incrementally.

The introduced method, I2RL-MEOM, approximates the unknown observation model using a shared feature set.  They then fit this model to observations with a max-entropy optimization (Equation 8).  The learned feature expectations are then used to incrementally update the IRL objective. The authors additionally demonstrate the convergence properties of the method.

The method is compared against AIRL on a variety of robotic manipulation tasks in an onion sorting domain and a perimeter breaching task.

**Issues:**

The main issue with this work is the extremely notational heavy exposition.  I believe that better summarization of the method is also needed, with only the most relevant equations concisely shown in the main text.  I also encourage the authors to examine more baselines for comparison outside of AIRL as stated in my main review.

**Quality Of The Limitations Section:**

Limitations are addressed clearly

**Reviewer Expertise:**

3: The reviewer is fairly confident that the evaluation is correct

**Robotics Focus:**

Highly relevant to robotics but no hardware experiments

**Strengths And Weaknesses:**

I2RL-MEOM is compared against AIRL on a variety of Sawyer robotic manipulation tasks in an onion sorting domain. Some onions are blemished and must be manipulated appropriately depending on the task (picked, sorted, etc). There are decent improvements over AIRL, but the results are not overtly impressive. The convergence analysis is also appreciated.

This method shows some decent improvements in a set of interesting and realistic Saywer sorting tasks. It also deals with a difficult and realistic problem, when the observation model is noisy in the IRL setting. There are decent improvements over AIRL, but the results are not overtly impressive.

Overall, I find this paper incremental, and limited. I believe that learning the observation model incrementally is too restrictive under this task, such as its restriction to binary features. I also believe that the experimental justification is lacking.

This paper is difficult to read at times, mostly since such a large amount of notation is introduced. For example, Section 5.1 is extremely dense with rapid-fire notation heavy descriptions of each algorithm step. I feel that this entire section could be replaced by a simplified pseudo-code figure. There are many descriptions in this paper where the notation could and should be simplified. Some formula-heavy related work could be summarized better (especially Section 3.1), with the full exposition in the appendix.

The results are poorly organized, Figure 3 (Right) is missing actual columns for the TP, FP, and result value, where instead, commas are used to separate results. Figure 4 is also too small. The exhaustive list of experiments (Lines 213-224) should also be moved to the appendix and only summarized in this section.

I am confused as to why the only baseline compared is AIRL. I feel that more extensive comparisons are needed against other model-based methods. It would be nice to see a comparison against Robust IRL, even with an incorrect observation model. The empirical justification for this method is, therefore, quite limited. I would also encourage the authors to compare against other SOTA Robust IRL methods such as Viano et al., 2021 (https://openreview.net/pdf?id=t8HduwpoQQv).

This method is also severely limited by the fact that features are binary functions. The authors rightfully explain this limitation in Section 7.



**Summary Of Recommendation:**

I still believe this work is quite restrictive, and can only deal with a subset of feature functions.  The experimental results are lacking and suffer from a lack of sufficient baseline comparisons.  The improvements over AIRL are decent, but not overtly impressive. The writing is very hard to work through and is too notation heavy.  The algorithm description therefore suffers.

---

> ### Author Response · Authors · 2022-08-27
> **Authors' Response**
>
> We see nothing restrictive about learning the observation model incrementally. We admit the  restriction to binary features at the end of the paper. We can potentially accommodate continuous valued  observation features by introducing a metric for the degree of corruption of ground truth due to noise, instead of whether/not (eqn 6). $P(\psi)$ would then be densities.
>
> Please note that there is no new notation in Sec. 5.1. All notations were introduced in prior sections. The full Algorithm was already listed in the Appendix. Page limitation and a single column format forced us to defer the algorithm (best suited for 2-column format) to the Appendix. Notations can admittedly get ugly due to the inherent complexity of the matter; consider for example the notation $\rho=\rho^{\pi^{soft}_{M^{train}_{\theta^{train}}}}_{M^{train}}$ in Sec. 3.3 of Viano et al., 2021 that you cited. However, we have tried to use reasonable, intuitive notations with careful consistency in the text.
>
> Notation heaviness in some places allowed us to be less verbose and meet the page limitation. We gratefully accept your suggestion of moving the exhaustive feature lists to the Appendix, to buy space to reduce notational clutter, and for a less formula-heavy exposition.
>
> We included AIRL as a baseline because we know from experience that if a GAN based baseline is not included then the reviewers would want to see a comparison with one. We are  now adding another model-based baseline – MLIRL [VRoman, M. C. 2014; Maximum Likelihood Inverse Reinforcement Learning. Ph.D. Dissertation, Rutgers University] – as you suggested. So far we have obtained the LBA comparison with MLIRL for just the perimeter patrol domain (please see the attachment); experiments in onion sorting is on-going. Before trying MLIRL, we tried a more recent algorithm – Scalable BIRL [ICLR 2021; https://github.com/XanderJC/scalable-birl] – but failed to run it due to JAX compatibility issues.
>
> Robust-IRL with an incorrect observation model – the baseline that you wanted us to add – is already included in the experimental comparison. RI2RL-FQOM is, in fact, the incremental variant of Robust-IRL with an incorrect observation model. We have also considered Viano et al., 2021, as suggested, for an additional baseline, but didn’t find it cogent to our topic. In that paper, Viano et al. consider an orthogonal problem of mismatch in the transition function between the learner’s and demonstrator’s tasks. In our setting, the two tasks are the same. So, there would be no mismatch, therefore that algorithm would boil down to vanilla MaxEnt-IRL – already the basis of our algorithm. When it comes to IRL methods that address observation noise, our choices on baselines are truly limited.

---

### Official Review · Reviewer_F2Vv · 2022-08-03

**Originality:** Very Good
**Technical Quality:** Excellent
**Clarity Of Presentation:** Good
**Impact:** 3

**Recommendation:**

Strong Accept: I recommend accepting the paper and will argue for my recommendation even if other reviewers hold a different opinion.

**Summary:**

This paper considers the problem of reward learning when the observation model is unknown. The authors propose a maximum entropy approach to estimate the observation model an incorporate this into an online IRL algorithm. Results on real robots show that the proposed approach outperforms AIRL and a frequentist version. Theoretical analysis shows that the proposed approach has desirable convergence properties.

**Issues:**

The algorithm abbreviation (RI2RL-MEOM) is quite long and awkward. I'd suggest something shorter and catchier :)

Figure 2 has a lot of clutter with all the equations/notation in the upper right. I would recommend using words or pictures to describe as much as possible as done in the left half of the figure.

How would the authors propose that the reward features and observation features be chosen? The method seems to rely heavily on these being good representations. Could they be learned from data?

Does AIRL receive the same observation representations as the proposed approach?

How critical is the SA-network? Would the proposed approach work on raw continuous states and actions?

I would be interested in seeing what is lost due to an unknown observation model. Is it possible to run prior Robust IRL in simulation with a known model and compare to see how much learning the observation model degrades performance?

**Quality Of The Limitations Section:**

Limitations are addressed clearly

**Reviewer Expertise:**

4: The reviewer is confident but not absolutely certain that the evaluation is correct

**Robotics Focus:**

Sufficient demonstration on hardware

**Strengths And Weaknesses:**

Strengths
- online reward learning with an unknown observation model is an important and understudied problem
- good overview of prior work in section 3
- nice theoretical analysis of proposed algorithm
- real robot experiments

Weaknesses
- Seems to require fully solving the MDP as does classical MaxEnt IRL
- Requires discrete and binary features and known reward features

**Summary Of Recommendation:**

This paper provides a nice mix of theory, technical rigor, and empirical results on physical robots. The experimental results show the advantage off the proposed approach and the overall problem studied by the authors seems novel and important for real world robot reward learning.

---

The authors have addresses my main concerns. I still think the paper should be accepted.

---

> ### Author Response · Authors · 2022-08-27
> **Authors' Response**
>
> We are currently discussing alternatives to the abbreviation RI2RL-MEOM that are concise. One suggestion that we are considering is: RIME ([R]obust [I]ncremental IRL with [M]ax [E]ntropy Observation Model). Meanwhile, we continue to use the previous acronyms during the rebuttal phase.
>
> We have redone Fig. 2. Please see the attachment.
>
> Our lead author selected the features by focusing on those aspects that contribute to the goal of an RL agent, viz., inspecting and distinguishing good/bad onions, and where they go. Similarly in the perimeter patrol domain, the key is where the patroller turns. Based on these considerations, a general suggestion for manual feature construction would be to think about the various decisions involved in the performance of the task, and codify the decisions as feature functions whose values depend on the value of the decision at each state. Sometimes, the learner may know certain features apriori, and would want to ascertain the demonstrator’s preferences in terms of those features. In such cases, methods that automatically construct/extract features from data are less useful. In other cases, a more appropriate technique might be to use a neural network to map the input images augmented with $c(s,a)$ values directly to reward values, which would obviate the need for manually constructed predicates. Here, rewards would just be a function of (s,a), and any automatically constructed intermediate feature would be hidden in the deep pipeline. Using neural network approximation, maximum entropy IRL can still be implemented by using backpropagation for updating the learned reward function (see additional citations [A,B] below).
>
> AIRL had the same input as the other algorithms, viz., the output of the perception process.
>
> Criticality of SA-Net: The proposed IRL algorithm does not work on continuous states and actions. So, a multi-label classifier such as SA-Net is needed to classify the streams of raw RGB-D data into state-action pairs.
>
> While it is certainly possible to run Robust-IRL with a known observation model in a toy problem, we believe the evidence at hand should suffice. We believe RI2RL-FQOM offers a good baseline that clearly demonstrates the need to learn a better, high-quality observation model. Furthermore, the previous paper on Robust IRL (Shahryari & Doshi, 2017) compares Robust IRL with a baseline that doesn’t learn the observation model, and improves significantly on it. It’s clear that learning an observation model helps with IRL. Lastly, even if the observation model was accurate, the resulting LBA couldn’t be higher than 1. Our method approaches 96% of full LBA, therefore any degradation in performance due to learning the observation model couldn’t possibly be large.
>
> [A] Finn, Chelsea and Levine, Sergey and Abbeel, Pieter: Guided Cost Learning: Deep Inverse Optimal Control via Policy Optimization. In arXiv preprint; arXiv:1603.00448. 2016
>
> [B] Wulfmeier, Markus and Posner, Ingmar: Maximum Entropy Deep Inverse Reinforcement Learning. In arXiv preprint. 2015.

---

### Official Review · Reviewer_Kim8 · 2022-08-05

**Originality:** Good
**Technical Quality:** Good
**Clarity Of Presentation:** Very Good
**Impact:** 3

**Recommendation:**

Weak Accept: I recommend accepting the paper, but will not argue for my recommendation if the majority of other reviewers have a different opinion.

**Summary:**

They propose to extend incremental/online inverse reinforcement learning (I2RL) to the case with noisy observations and an unknown observation model, making it suitable to real-world robotics applications. Their method -- called Robust Incremental IRL with Maximum Entropy Estimation of Observation Model (RI2RL-MEOM) -- is formulated as a sequence of incremental learning sessions where each session is a two-step process: (1) learning a maximum entropy distribution of the observation model, and (2) using that model to learn the reward function of the expert. They present results on 2 robotics tasks: (1) onion sorting with Sawyer arm based on human demonstration and (2) breaching a perimeter patrol with 2 Turtlebots. RI2RL-MEOM learns a more accurate policy compared to (a) an IRL method that doesn't learn an observation model, and (b) a custom baseline that learns a simpler observation model.

**Issues:**

I didn't find any major issues with this paper. However, I'm not familiar with the theoretical component of this work, and I defer the judgement on this to the other reviewers.

**Quality Of The Limitations Section:**

Limitations are addressed clearly

**Reviewer Expertise:**

2: The reviewer is willing to defend the evaluation, but it is quite likely that the reviewer did not understand central parts of the paper

**Robotics Focus:**

Highly relevant to robotics but no hardware experiments

**Strengths And Weaknesses:**

Strengths

* Their approach is practical for robotic applications, where the observations are typically noisy and the observation model is unknown.
* RI2RL-MEOM admits formal guarantees of monotonic convergence and sample complexity bound.
* Their experiment on onion sorting uses _real_ videos of human sorting onions, which is a realistic setting.
* Their experiment demonstrates low KL divergence for a simulated dataset of the onion sorting task, thus demonstrating the correctness of the learned model against a ground truth. (The simulated dataset is generated for the purposes of evaluating the learned observation model against a known model.)
* Their method achieves higher learned behavior accuracy (LBA) and requires less data (i.e. sessions) compared to their baselines -- RI2RL-FQOM and AIRL.

Weaknesses

* They required a manually defined set of features. Although this was appropriate for the tasks considered, this might be a limitation for more complex tasks that might not have obvious features or features that are non-trivial to extract.
* Although they did use real videos as training data for the onion sorting task, the evaluation of the learned policy was performed in simulation. However, this might be appropriate for their setting since they are testing for high-level decision making.

**Summary Of Recommendation:**

I believe this work is a good example of a learned-based method that addresses real-world constraints -- namely noisy observations and a unknown observation model -- and demonstrate results on a task that uses real-world data.

---

> ### Author Response · Authors · 2022-08-27
> **Authors' Response**
>
> Indeed, constructing features manually appears to be burdensome, and techniques such as deep learning that automatically construct features may be an attractive alternative. However, consider situations where the learner wants to ascertain the demonstrator’s preferences in terms of features it already knows, e.g., when an onion is in the eye/inspection region. It seems unlikely that a deep neural network will discover this exact feature, or even express it in an intelligible way.
>
> We also used two unrelated tasks to demonstrate that our method is sufficiently general for broader applicability.
>
> Please note that a part of the evaluation, specifically LBA, was also conducted with real data and not just in simulation. KLD evaluation could not be performed with real data as it would not allow us to control $P(\psi)$. We attest to your position that this is probably adequate for high-level decision making.

---

### Meta-Review · Area_Chair_6utc · 2022-08-13

**Recommendation:** Accept (Poster)
**Confidence:** 3

**Metareview:**

Overall, the problem of learning to imitate given noisy and unknown observations is relevant to the community. The approach is original although somewhat incremental, and the technical quality appears to be sound. It is great that real human demonstrations were used for the onion sorting task. Additionally, the inclusion of a formal theoretical analysis of the approach is a strength of the paper. However, there were a few weaknesses that should be addressed. First, the clarity of the paper could be improved by simplifying the notation, as multiple reviewers found the complex notation made the paper difficult to read. The use of binary/manual features seems to be a limiting factor, and the paper would also be improved by comparing to other approaches rather than only AIRL. The authors are encouraged to address these concerns along with other concerns raised by the reviewers.

========

Thank you to the authors for responding to the authors concerns. Overall, the authors address a relevant problem for the community and evaluate their approach on realistic experiments, which I believe warrants acceptance. I agree that needing to hand-specify features is a limiting factor of this approach, and the authors are encouraged to discuss how such features could be learnt within the text of the data. Additionally, the authors should aim to simplify notation within the paper to make it easier to read.

**Best Paper Nomination:**

No